# Hydrogen Diffusivity in Different Microstructures of 42CrMo4 Steel

**Atif Imdad, Alfredo Zafra, Victor Arniella and Javier Belzunce \***

University of Oviedo, Polytechnic School of Engineering of Gijón, East Building, Department of Materials Science and Metallurgical Engineering, 33203 Asturias, Spain; UO256361@uniovi.es (A.I.); zafraalfredo@uniovi.es (A.Z.); arniellavictor@uniovi.es (V.A.)

\* Correspondence: belzunce@uniovi.es

**Abstract:** It is well known that the presence of hydrogen decreases the mechanical properties of ferritic steels, giving rise to the phenomenon known as hydrogen embrittlement (HE). The sensitivity to HE increases with the strength of the steel due to the increase of its microstructural defects (hydrogen traps), which eventually increase hydrogen solubility and decrease hydrogen diffusivity in the steel. The aim of this work is to study hydrogen diffusivity in a 42CrMo4 steel submitted to different heat treatments—annealing, normalizing and quench and tempering—to obtain different microstructures, with a broad range of hardness levels. Electrochemical hydrogen permeation tests were performed in a modified Devanathan and Stachursky double-cell. The build-up transient methodology allowed the determination of the apparent hydrogen diffusion coefficient, $D_{app}$, and assessment of its evolution during the progressive filling of the microstructural hydrogen traps. Consequently, the lattice hydrogen diffusion coefficient, $D_L$, was determined. Optical and scanning electron microscopy (SEM) were employed to examine the steel microstructures in order to understand their interaction with hydrogen atoms. In general, the results show that the permeation parameters are strongly related to the steel hardness, being less affected by the type of microstructure.

**Keywords:** hydrogen diffusion; hydrogen permeation tests; heat treatments; microstructures; 42CrMo4 steel

## 1. Introduction

In a global context of environmental awareness, the utilization of hydrogen as an energy source has been the subject of intensive study in the last few years [1]. Due to its versatility and zero $CO_2$ emissions, hydrogen is expected to be a good choice for future energy systems [2,3]. Therefore, the study of economical materials that can safely store and transport hydrogen has become a topic of general interest. Specifically, medium and high-strength low-alloy steels have been used in numerous investigations in order to elucidate the influence of hydrogen on their mechanical properties [4].

Hydrogen can be absorbed by steel during fabrication (casting, electroplating, electrochemical machining, pickling, welding, etc.) and during its service (cathodic protection of offshore or buried structures, pipes, and vessels containing high pressure hydrogen gas or hydrogen containing gases, etc.) giving rise to the HE phenomenon [5]. A large number of catastrophic structural failures have been attributed to HE [6,7]. These kinds of failure occur under stress levels lower than the design stress; they are brittle in appearance—with minimum previous plastic deformation—and hence very difficult to predict. Accordingly, steel structures that work in the presence of this element must be properly designed with steels able to withstand HE. In particular, it must be borne in mind that the deterioration of mechanical properties—reduction in strength, fracture toughness and ductility, and crack growth rate enhancement [8,9]—due to the presence of internal hydrogen, increases with the strength of the steel [10].

Hydrogen diffusion and trapping in steels have been studied in order to understand the HE process. Hydrogen atoms have a high mobility in BCC steel lattices, which implies that they can easily interact with their microstructural defects, where they can be temporarily or permanently trapped [11]. Typical hydrogen traps in steels are vacancies, microvoids, dislocations, grain boundaries, matrix-carbides interfaces, and other internal interfaces [12]. These traps can be classified as reversible and irreversible depending on their ability to retain hydrogen at room temperature (RT). Irreversible traps have a high trap activation energy, therefore the hydrogen trapped in them behaves as non-diffusible at RT. In contrast, sites with lower trap activation energy constitute reversible traps, which simply delay hydrogen diffusion at RT [13]. Many studies have employed temperature-programmed desorption to characterize hydrogen trapping in steels [12,14–17].

Microstructural traps delay hydrogen diffusivity, increasing the residence time of this element in the steel and so the risk of HE. Among all the available techniques to characterize hydrogen diffusivity in steels, electrochemical hydrogen permeation has proven to be one of the most reliable methods [18–21].

For instance, Parvathavarthini et al. [22] studied hydrogen diffusion in a 2.25Cr1Mo steel submitted to a variety of heat treatments, reporting an inverse correlation between steel hardness and hydrogen diffusivity. This is in line with the findings of other authors, such as Moli-Sanchez et al. [23] and Depover et al. [15]. In general, they agree that the dislocation density reduction that takes place during certain heat treatments—high temperature tempering or annealing—is responsible for decreasing the density of hydrogen traps (mainly dislocations) and thus increasing hydrogen diffusivity. Galindo-Nava et al. [24] also demonstrated that dislocations are the main factor in controlling the mobility of hydrogen in quenched and tempered martensitic steels.

Other researchers have studied the influence of microstructure and hardness on hydrogen transport and trapping in steels [25–28]. For example, L.B. Peral et al. [29] studied the interaction between hydrogen atoms and the microstructure of quenched and tempered 2.25Cr1Mo and 2.25Cr1MoV steels, underlaying the influence of the steel hardness and of the addition of small amounts of vanadium in hydrogen diffusivity. Chan [30] studied hydrogen diffusivity and trapping in steels with different microstructures. They found that hydrogen pick up decreases with the austenite transformation temperature and, furthermore, not only grain size but also the nature of the grain boundary determines the hydrogen trapping ability and diffusivity. Thomas and Szpunar [31] observed a decrease of the diffusion coefficient and hydrogen trapping with the increase of the grain size in an X-70 ferrite-pearlite pipeline steel. Finally, according to the work of Nanninga et al. [32] on steels with different carbon contents and microstructures, hardness superseded the effects of microstructure and alloying in governing the susceptibility to HE.

Regarding the 42CrMo4 steel used in this study, although Zafra et al. [17,33] already assessed the influence of the grain size, the plastic deformation, and the tempering temperature in the hydrogen permeation parameters, other aspects, such as the influence of the tempering time on quenching and tempering treatments or the austenitization temperature in normalizing and annealing treatments, still need to be thoroughly examined.

Consequently, electrochemical hydrogen permeation tests have been performed in this work in order to study hydrogen diffusivity in a 42CrMo4 steel. Different microstructures were produced by means of annealing, normalizing, and quench and tempering heat treatments. The influence of the hardness, the austenitization temperature and the tempering time on both the apparent and the lattice hydrogen diffusion coefficients was assessed.

## 2. Materials and Methods

### 2.1. Steel and Heat Treatments

The material studied in this work was a 42CrMo4 steel. Table 1 shows the chemical composition of the steel.

**Table 1.** Chemical composition (weight %) of the 42CrMo4 steel used in this study.

| %C | %Cr | %Mo | %Mn | %Si | %S | %P |
|-----|------|------|------|------|-------|-------|
| 0.42 | 0.98 | 0.22 | 0.62 | 0.18 | 0.002 | 0.008 |

In order to analyze the influence of the steel microstructure on its hydrogen permeation behavior, different heat treatments were carried out onto hot rolled 250 mm × 125 mm × 12 mm plates. The nomenclature of the obtained 42CrMo4 steel grades and their correspondent heat treatments are shown in Table 2.

**Table 2.** Heat treatments applied to 42CrMo4 steel and obtained grades.

| Steel Grade | Heat Treatment |
|-------------|----------------|
| QT600-3min | 845 °C/40 min + water quench + 600 °C/3 min tempering |
| QT600-30min | 845 °C/40 min + water quench + 600 °C/30 min tempering |
| QT600-2h | 845 °C/40 min + water quench + 600 °C/2 h tempering |
| QT600-24h | 845 °C/40 min + water quench + 600 °C/24 h tempering |
| QT600-7d | 845 °C/40 min + water quench + 600 °C/7 days tempering |
| QT725-4h | 845 °C/40 min + water quench + 725 °C/4 h tempering |
| 845FC | 845 °C/40 min + furnace cooling |
| 1050FC | 1050 °C/40 min + furnace cooling |
| 845AC | 845 °C/40 min + air cooling |
| 1050AC | 1050 °C/40 min + air cooing |

### 2.2. Hardness and Microstructural Characterization

Small, handy specimens machined from the heat-treated plates were firstly ground with SiC papers of different grit sizes (from #60 to #1200) and then successively polished in synthetic cloths with 6 µm and 1 µm diamond paste. After completing the polishing process, samples were etched with Nital-2% and their microstructures observed using a Nikon optical microscope ECLIPSE MA200 (Material Science Laboratory, University of Oviedo, Gijón, Spain) and a JEOL scanning electron microscope JSM5600 (Scientific and Technical Services, University of Oviedo, Gijón, Spain). Vickers hardness (HV30) measurements were performed applying a load of 30 kg for 15 s.

### 2.3. Hydrogen Permeation Tests

Hydrogen permeation tests were performed on all the heat-treated grades shown in Table 2 in order to determine the hydrogen diffusion and trapping behavior. Specimens of 30 mm × 25 mm were ground up to #1200 SiC paper until a final thickness between 0.7 and 1 mm. They were cleaned with water and acetone before starting the test.

The permeation tests were carried out in a double electrolytic cell based on the one developed by Devanathan and Stachurski [34], which is schematically shown in Figure 1a.

Hydrogen generation occurs in the cathodic cell, where hydrogen reduction takes place ($2H^+ + 2e^- \rightarrow 2H_{ads}$, Figure 1b). Hydrogen atoms are adsorbed ($H_{ads}$) onto the steel surface and then absorbed into the steel ($H_{abs}$). This cell was filled with 300 mL of an acid solution (pH ≈ 1) composed of 1 M $H_2SO_4$ and 0.25 g/L $As_2O_3$. In this case, $H_2SO_4$ is the hydrogen donor and $As_2O_3$ is added to limit the hydrogen recombination reaction ($H_{ads} + H_{ads} \rightarrow H_2$) [35], which considerably decreases the efficiency of the permeation process [33]. The anodic cell, where hydrogen is desorbed ($H_{des}$) from the specimen and the oxidation reaction occurs ($H_{des} \rightarrow H^+ + e^-$, Figure 1b), was filled with 300 mL of a basic solution (pH ≈ 12) of 0.1 M NaOH. The cells are separated by the specimen, which represents the working electrode (WE) in each cell. A circular exposed area of 1.25 cm² was always used. Hydrogen oxidation was enhanced in the anodic cell, ensuring a virtually zero hydrogen concentration on the exit side of the specimen (Figure 1b) via the electrodeposition of a thin palladium layer. It also ensures that only hydrogen oxidation is taking place in the anodic cell [36]. This coating, with a thickness of approximately 50 nm,

measured using SEM, was electrochemically deposited from a commercial palladium solution of 2 g/L of Pd, applying a current density of 1 mA/cm$^2$ for 5 min [37].

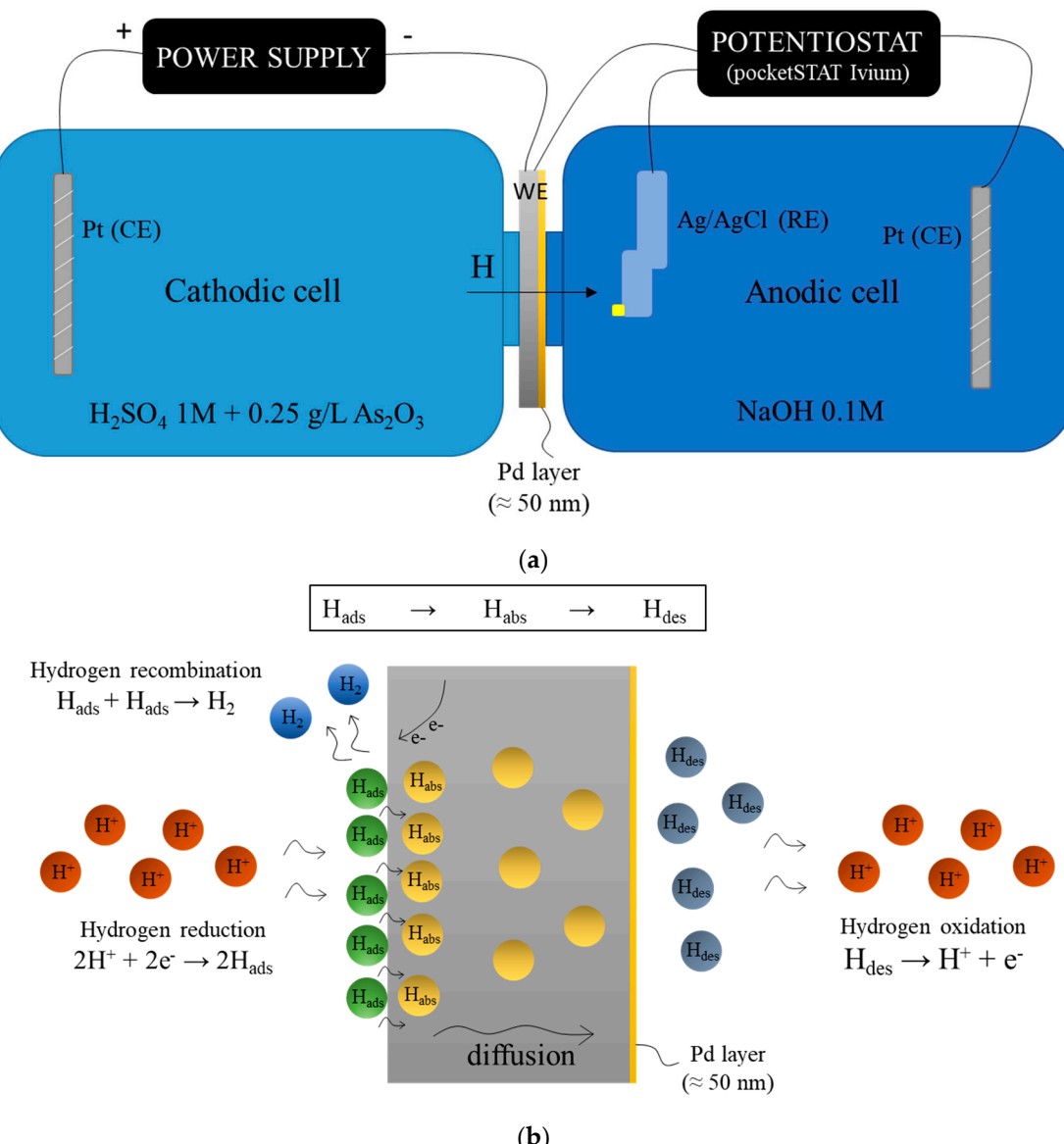

**Figure 1.** (**a**) Scheme of the permeation double-cell employed in this work and (**b**) all the steps taking place in the hydrogen permeation process, from hydrogen reduction in the cathodic cell (left) to hydrogen oxidation in the anodic cell (right).

Thin platinum plates with a total surface area of 1 cm$^2$ were used as counter electrodes (CE) in each cell. A reference silver/silver chloride electrode (Ag/AgCl, RE) was also employed in the anodic cell to keep the sample at a constant anodic potential of $\approx -50$ mV (open-circuit potential) along all the permeation test. Before starting the tests, the background current density in the anodic cell may be below 0.1 $\mu$A/cm$^2$, which in any case was subtracted from the measured oxidation current prior to analysis. The oxidation or permeation current density, $J_p$, was continuously recorded in the anodic surface of the specimen by means of a pocketSTAT Ivium potentiostat with a current operation range of $\pm 10$ mA. All tests were performed at RT.

Successive Build-Up Permeation Transients

The permeation method employed in this study consisted in sequentially increasing the applied cathodic current density, $J_c$, and recording the associated build-up permeation

transients, as can be observed in Figure 2a. Operating this way, it is possible to determine a value of $D_{app}$ for each transient and establish the relationship between $D_{app}$ and $J_c$. During these tests, the microstructural traps are progressively filled with hydrogen. Once saturation is reached, the hydrogen diffusion coefficient stabilizes, which means that diffusivity is no longer affected by hydrogen trapping. This is known as the lattice diffusion coefficient of the steel, $D_L$. Some authors have successfully applied this methodology to characterize hydrogen diffusivity in CrMo steels [33,38–40].

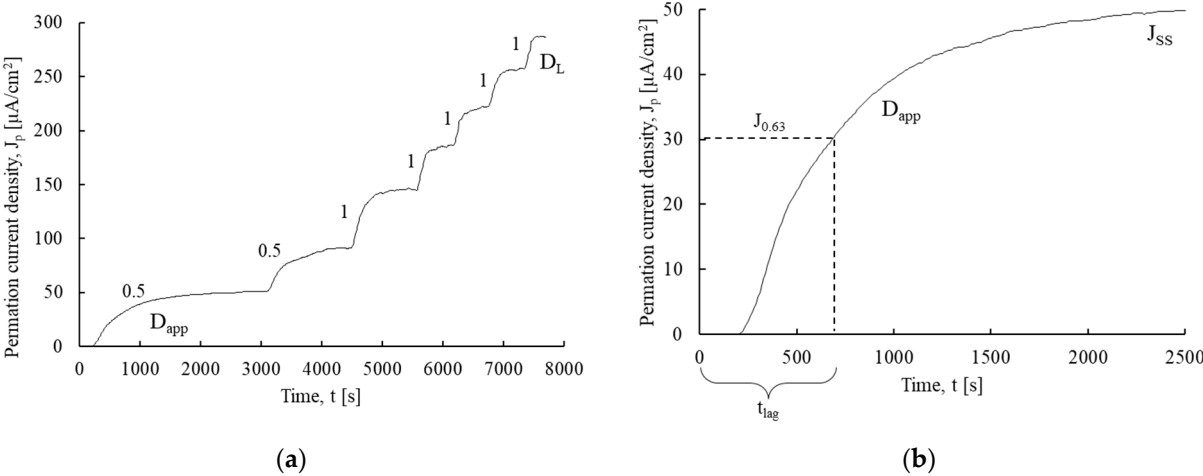

**Figure 2.** (**a**) Characteristic build-up permeation transients produced increasing the cathodic current density ($J_c$ = 0.5 + 0.5 + 1 + 1 + 1 + 1 + 1 mA/cm$^2$) and (**b**) example of $D_{app}$ calculation in a particular transient.

The cathodic current density was increased in steps of 0.5 mA/cm$^2$ for the first two transients, and of 1 mA/cm$^2$ for the following ones (see Figure 2a). As shown in Figure 2b, $D_{app}$ was calculated at each transient following the lag time method [41]. Equation (1), derived from Fick's diffusion solution under the appropriate boundary conditions, was used for $D_{app}$ determination:

$$D_{app} = \frac{L^2}{6 \cdot t_{lag}} \tag{1}$$

where $L$ is the specimen thickness, and $t_{lag}$ is the time needed to reach the 63% of the steady-state permeation current state, $J_{ss}$. The first permeation transient performed under a cathodic current density of 0.5 mA/cm$^2$ allows us to determine $D_{app}$ under a very low hydrogen trap occupancy condition.

## 3. Results

### 3.1. Microstructural Characterization

#### 3.1.1. Quenched and Tempered 42CrMo4 Steel

The SEM microstructures of the six quenched and tempered 42CrMo steel grades are shown in Figure 3 under a magnification of 5000×. In addition, the HV30 (average ± standard deviation) measured on each steel grade is provided in Table 3.

The microstructure of all these 42CrMo4 grades was tempered martensite except for QT600-3min grade (Figure 3a), in which, due to the extremely short duration of the tempering treatment, it was basically untempered martensite, with a high hardness of 484 HV30. As for the rest of the treatments performed at 600 °C, it is worth noting that tempering time is inversely related to the microstructure acicularity and thus to the distortion of the martensitic structure (dislocation density) [42]. It is also observed that carbide morphology and size are strongly related to the tempering time. Elongated carbides precipitated first along grain and martensitic lath/pack/block boundaries (QT600-30min), but as the tempering time increases, these carbides break up, grow, globulize, and distribute more homogeneously (see for example QT600-7d). In line with these microstructural

changes, the hardness of the steel decreases considerably with the tempering duration, as recorded in Table 3. Furthermore, the greatest effect of tempering was produced in the steel grade tempered at 725 °C for 4 h, as observed in Figure 3f. Dislocation density and internal stress levels are certainly the lowest of all the Q + T steel grades, as it presents the lowest hardness, 206 HV30.

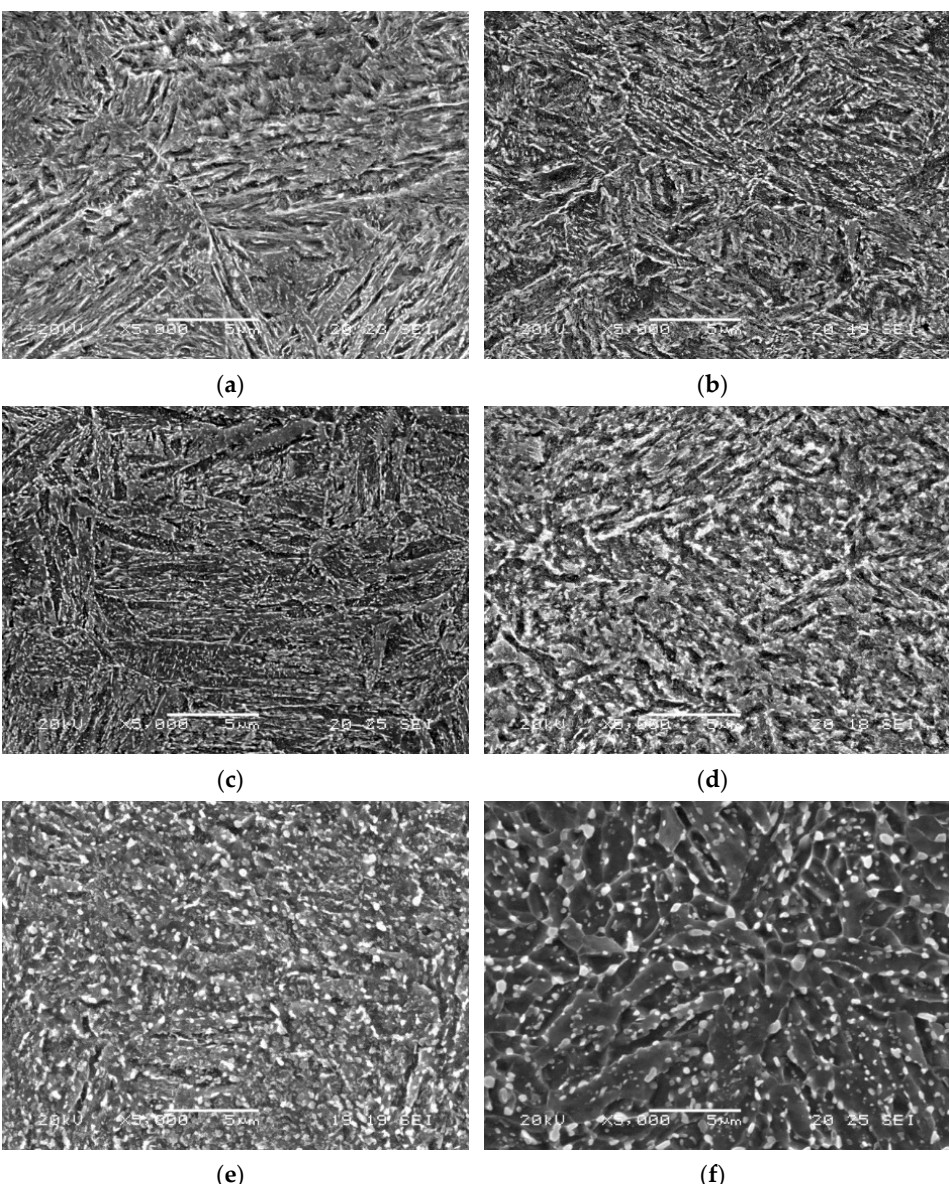

**Figure 3.** Microstructures (SEM, 5000×) of Q + T 42CrMo4 steel grades. (**a**) QT300-3min, (**b**) QT600-30min, (**c**) QT600-2h, (**d**) QT600-24 h, (**e**) QT600-7d, and (**f**) QT725-4h.

**Table 3.** Chemical composition (weight %) of the 42CrMo4 steel used in this study.

| Steel Grade | Microstructure | Hardness, HV30 |
|---|---|---|
| QT600-3min | Untempered martensite | 484 ± 3 |
| QT600-30min | Tempered martensite | 332 ± 4 |
| QT600-2h | Tempered martensite | 307 ± 2 |
| QT600-24h | Tempered martensite | 280 ± 7 |
| QT600-7d | Tempered martensite | 244 ± 5 |
| QT725-4h | Tempered martensite | 206 ± 3 |

### 3.1.2. Annealed and Normalized 42CrMo4 Steel

Figure 4 shows the optical and SEM micrographs of annealed (furnace cooled, FC) 42CrMo4 steel grades under different magnifications. A ferrite-pearlite microstructure is observed in both annealed grades. However, the grade annealed at lower temperature (845 °C) has a banded ferrite-pearlite microstructure, which was lost when the annealing was performed at 1050 °C. In the latter, a larger prior austenitic grain size is observed with ferrite precipitated along prior austenite grain boundaries. The slightly lower hardness measured in the steel austenitized at 845 °C (Table 4) is justified by the presence of a higher fraction of ferrite.

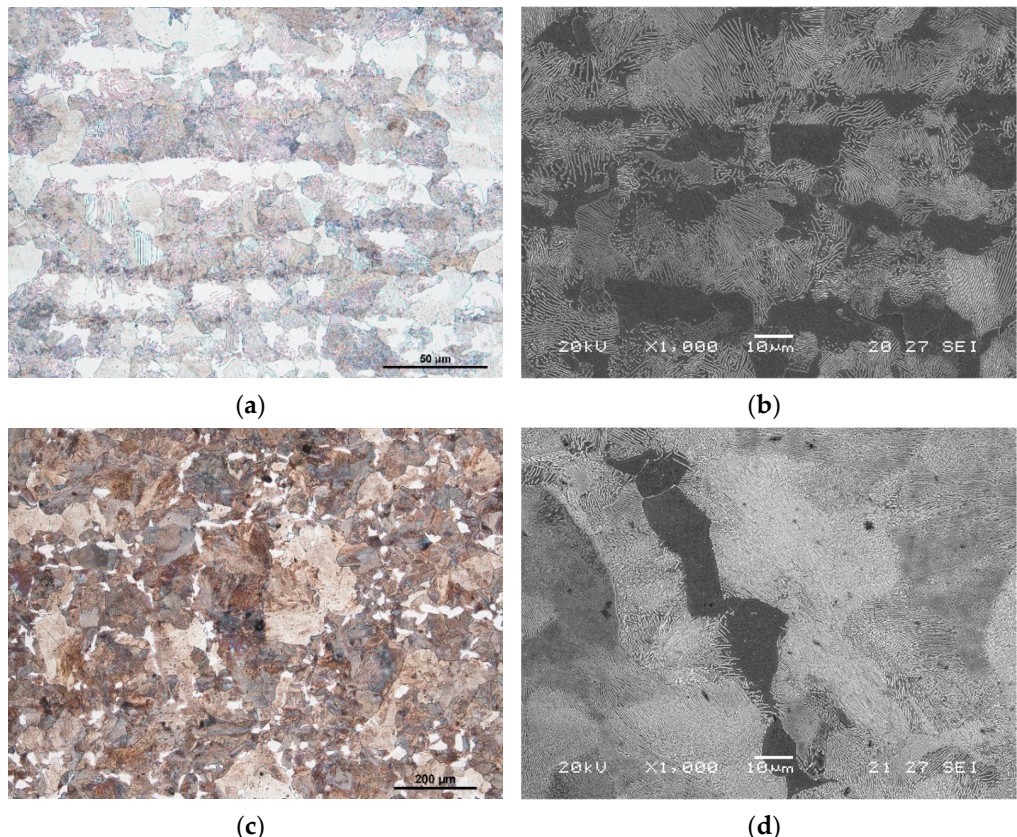

(a)  (b)

(c)  (d)

**Figure 4.** Microstructures of annealed 42CrMo4 steel: FC-845, (**a**) optical micrograph and (**b**) SEM micrograph; FC-1050, (**c**) optical micrograph and (**d**) SEM micrograph.

**Table 4.** Chemical composition (weight %) of the 42CrMo4 steel used in this study.

| Steel Grade | Microstructure | Hardness, HV30 |
|---|---|---|
| FC845 | Banded ferrite-pearlite | $183 \pm 6$ |
| FC1050 | Ferrite-pearlite | $210 \pm 3$ |
| AC845 | Bainite-ferrite-pearlite | $301 \pm 3$ |
| AC1050 | Bainite-ferrite-pearlite | $285 \pm 8$ |

The SEM microstructures of the 42CrMo4 steel after the normalizing treatments (air cooled, AC) are presented in Figure 5 under 1000× and 3000×. Both grades have complex microstructures, composed by bainite with some fractions of fine pearlite and only traces of ferrite, which conferred these grades a significantly greater hardness in comparison to the annealed ones (see Table 4). As can be clearly observed at 3000× (Figure 5d) the AC-1050 grade also presents a coarser microstructure due to the higher austenitization temperature. The presence of fewer internal interfaces justifies its lower hardness values.

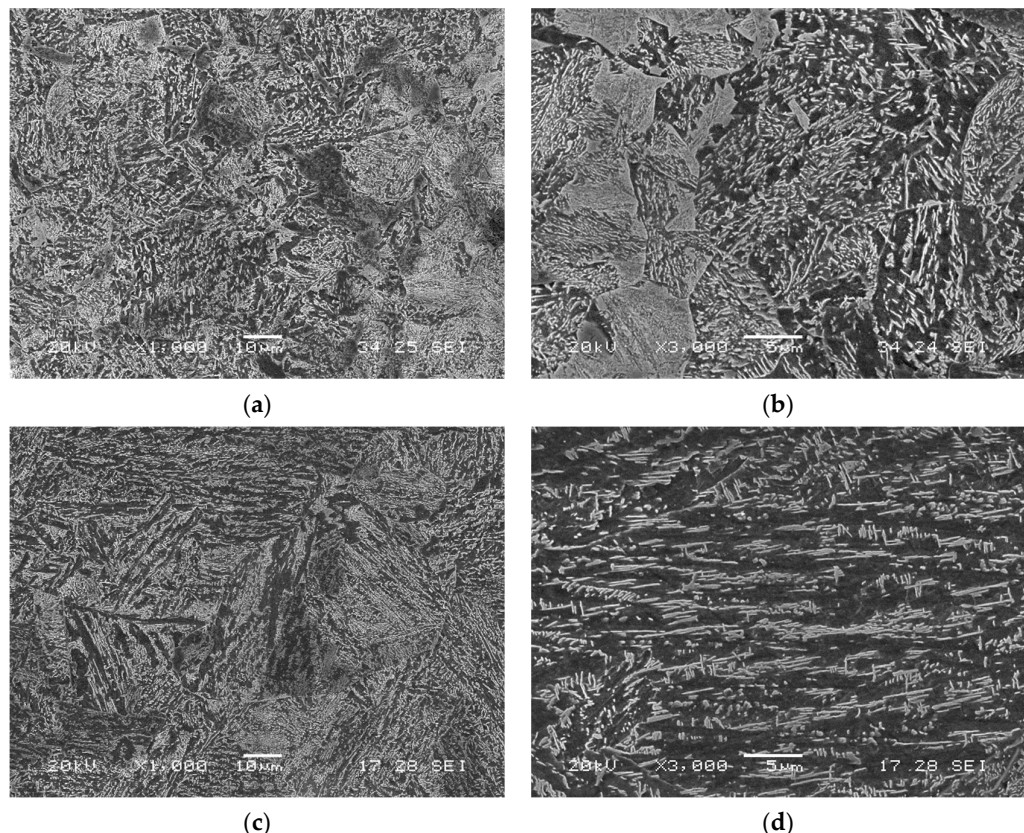

**Figure 5.** Microstructures (SEM) of normalized 42CrMo4 steel: AC-845, (**a**) 1000× and (**b**) 3000×; AC-1050, (**c**) 1000× and (**d**) 3000×.

## 3.2. Hydrogen Diffusion

### 3.2.1. Quenched and Tempered 42CrMo4 Steel

As an example, Figure 6 and Table 5 show the results of the permeation tests performed on the QT700-7d steel.

Figure 6 presents the $J_p$ registered on all the recorded stepped permeation transients. The different transients were produced by sequentially increasing $J_c$ from 0.5 mA/cm$^2$ up to a final value of 6–7 mA/cm$^2$. Table 5 shows the values of $J_c$, $J_{ss}$, $t_{lag}$, and the $D_{app}$ value calculated in each transient by means of Equation (1).

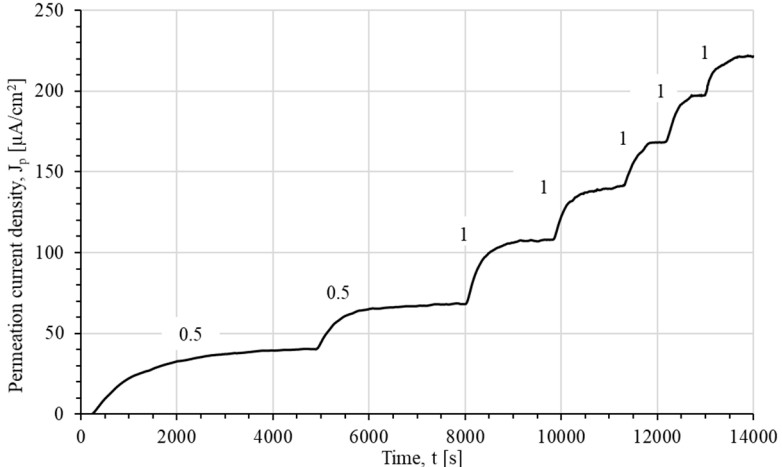

**Figure 6.** Evolution of the permeation current density over time for QT700-7 d grade ($J_c$ = 0.5 + 0.5 + 1 + 1 + 1 + 1 + 1 mA/cm$^2$).

**Table 5.** Results obtained in the stepped permeation tests carried out on the QT700-7d grade.

| Transient | $J_c$ [mA/cm$^2$] | $J_{ss}$ [µA/cm$^2$] | $t_{lag}$ [s] | $D_{app}$ [m$^2$/s] |
|-----------|-------------------|----------------------|---------------|---------------------|
| 1 | 0.5 | 40.3 | 1135 | $1.27 \times 10^{-10}$ |
| 2 | 1.0 | 68.2 | 495 | $2.91 \times 10^{-10}$ |
| 3 | 2.0 | 108.0 | 342 | $4.21 \times 10^{-10}$ |
| 4 | 3.0 | 141.4 | 297 | $4.85 \times 10^{-10}$ |
| 5 | 4.0 | 168.3 | 275 | $5.24 \times 10^{-10}$ |
| 6 | 5.0 | 197.4 | 233 | $6.19 \times 10^{-10}$ |
| 7 | 6.0 | 221.7 | 235 | $6.13 \times 10^{-10}$ |

It was observed that $J_{ss}$ increased proportionally with the increase of $J_c$, as more hydrogen was introduced in the steel microstructure. It is also worth noting that the $D_{app}$ value of the first transient was always the lowest ($1.27 \times 10^{-10}$ m$^2$/s) as all hydrogen microstructural traps are initially empty. In the following transients, hydrogen was progressively retained in the traps, increasing $D_{app}$ until a maximum and approximately constant value was attained in the last transients, $D_L$ (6.19–6.13 $\times 10^{-10}$ m$^2$/s). All the hydrogen microstructural traps present in the steel were saturated with hydrogen at this point [39]. These results are also in accordance with the theoretical work performed by Raina et al., which distinguished three different diffusion regimes depending on trap occupancy (traps empty, traps progressively filled, and saturated traps) [20].

The results obtained on all the Q + T grades are summarized in Figure 7, which represents the evolution of $D_{app}$ with $J_c$. Table 6 shows the values of $D_{app}$, calculated in the first transient (most traps empty) and $D_L$, obtained in the last transients (filled traps).

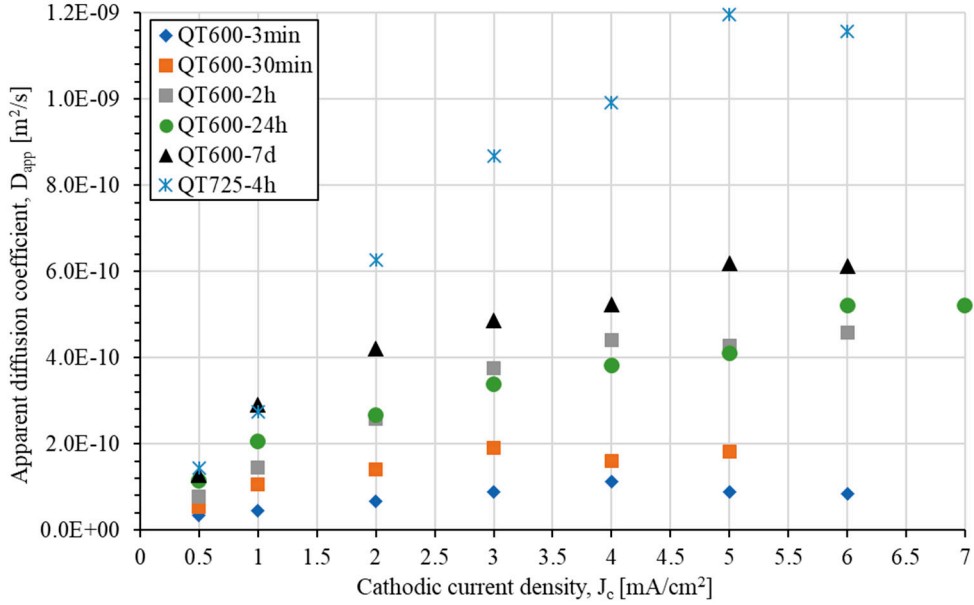

**Figure 7.** Evolution of $D_{app}$ with $J_c$ for the quenched and tempered 42CrMo4 grades.

**Table 6.** $D_{app}$ (first transient) and $D_L$ of quenched and tempered 42CrMo4 grades.

| Steel Grade | $D_{app}$ [m$^2$/s] | $D_L$ [m$^2$/s] |
|-------------|---------------------|------------------|
| QT600-3min | $3.43 \times 10^{-11}$ | $8.78 \times 10^{-11}$ |
| QT600-30min | $5.42 \times 10^{-11}$ | $1.61 \times 10^{-10}$ |
| QT600-2h | $7.71 \times 10^{-11}$ | $4.27 \times 10^{-10}$ |
| QT600-24h | $1.15 \times 10^{-10}$ | $5.22 \times 10^{-10}$ |
| QT600-7d | $1.27 \times 10^{-10}$ | $6.19 \times 10^{-10}$ |
| QT725-4h | $1.43 \times 10^{-10}$ | $1.16 \times 10^{-9}$ |

It is clearly appreciated that the shorter the tempering time (higher hardness), the lower the value of $D_{app}$ for any given $J_c$ level, which means that the density of hydrogen traps increases with the hardness of the steel. $D_L$ also increases with the tempering time, as the lattice distortion is progressively reduced during tempering [42]. Zafra et al. observed the same behavior in 42CrMo4 steel when increasing the tempering temperature while keeping the treatment time constant [17].

The grade tempered at 725 °C for 4 h after quenching has the highest $D_{app}$ and $D_L$ values. This behavior indicates that this grade has the lowest lattice distortion and dislocation density and thus the lowest density of hydrogen traps. This is in line with the microstructural recovery (Figure 3f) and low hardness level (Table 3) already discussed.

### 3.2.2. Annealed and Normalized 42CrMo4 Steel

Similarly, the $D_{app}$ vs. $J_c$ plot as well as the $D_{app}$ (first transient) and $D_L$ values obtained with the annealed and normalized 42CrMo4 grades are displayed in Figure 8 and Table 7, respectively. Again, $J_{ss}$ progressively increases with the increase of $J_c$, the $D_{app}$ value of the first transient was always the lowest, increasing until a maximum and approximately constant value was attained in the last transients.

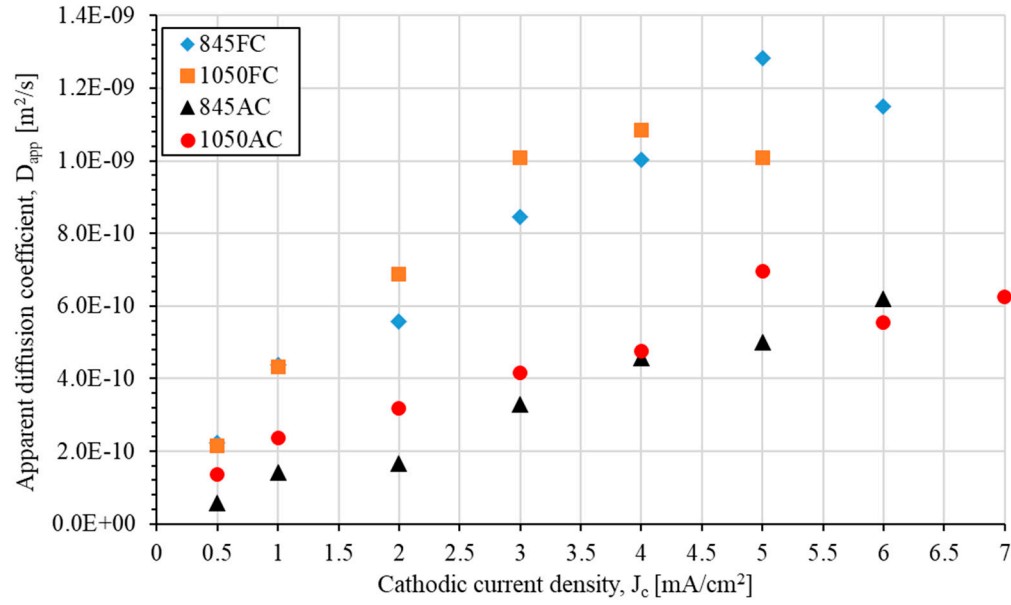

**Figure 8.** Evolution of $D_{app}$ with $J_c$ for the annealed and normalized 42CrMo4 grades.

**Table 7.** $D_{app}$ (first transient) and $D_L$ of quenched and tempered 42CrMo4 grades.

| Steel Grade | $D_{app}$ [m²/s] | $D_L$ [m²/s] |
|:---:|:---:|:---:|
| FC845 | $2.24 \times 10^{-10}$ | $1.15 \times 10^{-9}$ |
| FC1050 | $2.16 \times 10^{-10}$ | $1.01 \times 10^{-9}$ |
| AC845 | $5.63 \times 10^{-11}$ | $6.19 \times 10^{-10}$ |
| AC1050 | $1.36 \times 10^{-10}$ | $6.24 \times 10^{-10}$ |

Ferrite-pearlite microstructures (FC) always showed higher hydrogen diffusion coefficients than bainitic ones (AC), due to their lower hardness and thus dislocation density.

The effect of modifying the austenitizing temperature in the annealed microstructures was minimal, being the $D_{app}$ and $D_L$ values practically the same, even though the steel austenitized at 845 °C had a hardness slightly lower than that austenitized at 1050 °C.

On the other hand, a great variation in the first $D_{app}$ was detected between the two normalized grades. The finer microstructure—austenitized at 845 °C—displayed a greater density of internal interfaces and a slightly higher hardness having therefore a greater

number of hydrogen traps. However, this difference in diffusivity was progressively reduced as the hydrogen concentration in the microstructure (i.e., cathodic current density) increased and, eventually, a very similar $D_L$ was obtained in these two grades.

## 4. Discussion

Figure 9a,b respectively show the relationship between the steel hardness, HV30, the apparent hydrogen diffusion coefficient for the first transient, $D_{app}$, and the lattice diffusion coefficient, $D_L$. In general, a good correlation between these two hydrogen diffusion coefficients and the steel hardness is noticed.

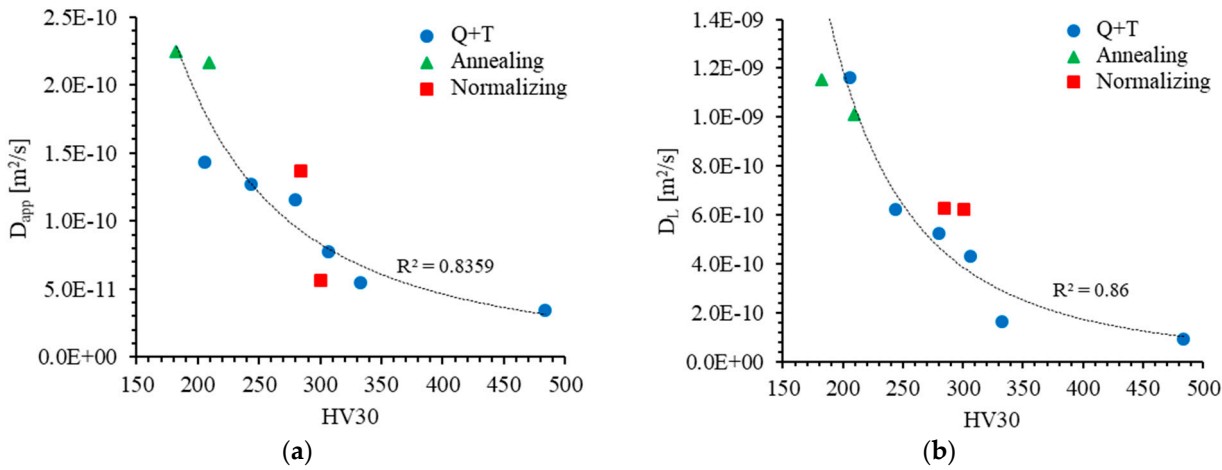

**Figure 9.** Relationship between hydrogen diffusivity and steel hardness. (**a**) $D_{app}$ of the first transient vs. HV30 and (**b**) $D_L$ vs. HV30.

First, it is observed that the steel microstructure (martensite with different degrees of tempering, ferrite + pearlite or bainite) barely affects $D_{app}$ and $D_L$ values. On the contrary, there exists a correlation between hardness and hydrogen diffusion coefficients of the steel. It is well known that hardness is directly related to the dislocation density present in the steel, since these linear structural defects are considered the most important microstructural traps present in martensitic CrMo steels [17,21,43]. The same situation is observed in the annealed (ferrite-pearlite) grades, although only two steels of this family were studied. Even though the steel annealed from 845 °C shows a fine and banded ferrite-pearlite microstructure, which was lost when the annealing was performed at 1050 °C (coarse microstructure with small fractions of ferrite precipitated along prior austenite grain boundaries), both grades display very similar $D_{app}$ and $D_L$ values. It should be mentioned here that Lee and Chan [44] reported similar hydrogen diffusivities (0.65, 1.59, and 1.68 × $10^{-10}$ m$^2$/s) for the through-thickness, longitudinal (rolling) and traverse directions of a 30CrMo4 steel with a banded ferrite-pearlite microstructure (achieved by heating to 870 °C for 1h and then furnace cooling). Therefore, the influence of ferrite-pearlite alignment in FC845 steel can be considered negligible in this work.

It is also interesting to notice that the $D_L$ obtained in the softest—more recovered—microstructures (QT725-4h, FC845 and FC1050), which lay between 1.01 and 1.16 × $10^{-9}$ m$^2$/s, is not far from the diffusivity reported by other authors for pure iron [11,45]. It is clear then that trapping phenomena is minimum in this situation, with lattice diffusion being predominant.

On the other hand, normalized steels with bainitic microstructures do not seem to follow exactly the same trend. In this case $D_{app}$ values are certainly affected by the steel microstructure: the coarsest bainite microstructure (AC1050), austenitized at a higher temperature, provides a considerably larger $D_{app}$ value (more than 2.4 times). Assuming that hydrogen atoms mainly diffuse through the bainitic-ferrite phase [46,47], a finer bainitic microstructure with higher density of internal interfaces gives rise to more tortuous diffusion paths, in which translates into lower hydrogen diffusion coefficients. However,

and regardless of their differences in hardness, the $D_L$ of both bainitic microstructures is identical. When most microstructural traps are filled, the possible differences in hydrogen diffusivity may be mainly attributed to internal structural distortion and residual stresses. It is then evident that after air cooling, both characteristics attain similar values in the two bainitic microstructures. In fact, when compared to the quenched and tempered microstructures, which were cooled very fast in water after austenitizing, the normalized steels have slightly higher $D_L$ values for similar hardness levels. Air-cooled bainitic microstructures, obtained under a relatively low cooling rate, are more relaxed and less distorted than the corresponding tempered martensite ones.

## 5. Conclusions

The present work studies the influence of the steel microstructure on hydrogen permeation by means of partial build-up permeation transient tests. Annealing, normalizing, and quench and tempering heat treatments were applied to hot rolled plates of a 42CrMo4 steel in order to obtain ferrite-pearlite, bainite, and tempered martensite microstructures, respectively.

The apparent hydrogen diffusion coefficient, $D_{app}$, associated with the first permeation transient was always the lowest in all the studied grades as most hydrogen microstructural traps were initially empty. As hydrogen atoms permeated the steel, $D_{app}$ increased due to the filling of the microstructural traps. A maximum and constant characteristic $D_{app}$ value, known as the lattice diffusion coefficient $D_L$, was reached in the last transients, when most hydrogen traps were practically saturated with hydrogen.

$D_{app}$ and $D_L$ values are strongly influenced by the hardness of the steel, with the kind of microstructure having a much lower influence. Only in the case of bainitic microstructures did the coarsest bainite give a larger $D_{app}$ value when most hydrogen microstructural traps were empty, which was justified because the hydrogen diffusion path through the bainitic ferrite was in this case less tortuous. On the other hand, differences in the lattice hydrogen diffusion coefficient, $D_L$, may be mainly attributed to internal structural distortion and residual stresses, as in this case most traps were filled.

**Author Contributions:** Conceptualization, A.Z. and J.B.; methodology, A.Z.; validation, J.B. and A.Z.; formal analysis, A.I. and A.Z.; investigation, A.I., A.Z. and V.A.; resources, J.B.; data curation, A.I. and A.Z.; writing—original draft preparation, A.I., A.Z. and V.A.; writing—review and editing, J.B. and A.Z.; visualization, A.I., A.Z. and V.A.; supervision, J.B.; project administration, J.B.; funding acquisition, J.B. All authors have read and agreed to the published version of the manuscript.

**Funding:** This research was funded by the Spanish Ministry of Science, Innovation and Universities, grant number RTI2018-096070-B-C31 (H2steelweld). A. Zafra acknowledges the financial support of the regional government of the Principality of Asturias for the Severo Ochoa grant PA-18-PF-BP17-038. V. Arniella thanks the funding from the Spanish State Research Agency with reference RTI2018-096070-B-C31.

**Data Availability Statement:** Most of the data is contained within the article. The curves obtained in all the permeation tests can be shared under demand.

**Acknowledgments:** The authors would like to acknowledge the technical support provided by the Scientific and Technical Service of the University of Oviedo for the use of the SEM JEOLJSM5600 scanning electron microscope.

**Conflicts of Interest:** The authors declare no conflict of interest.

## Abbreviations

The following symbols and abbreviations are used in this manuscript:

| | |
|---|---|
| AC | Air cooled |
| BCC | Body centered cubic |
| CE | Counter electrode |
| $D_{app}$ | Apparent hydrogen diffusion coefficient |
| $D_L$ | Lattice hydrogen diffusion coefficient |

| FC | Furnace cooled |
|---|---|
| $H_{abs}$ | Hydrogen absorbed in the steel |
| $H_{ads}$ | Hydrogen adsorbed on the cathodic surface of the specimen |
| $H_{des}$ | Hydrogen desorbed from the anodic surface of the specimen |
| HE | Hydrogen embrittlement |
| HV30 | Vickers hardness |
| $J_c$ | Cathodic current density |
| $J_p$ | Oxidation or permeation current density |
| $J_{ss}$ | Steady-state permeation current density |
| $L$ | Specimen thickness |
| Q + T | Quench and tempered |
| RE | Reference electrode |
| RT | Room temperature |
| SEM | Scanning electron microscope/microscopy |
| $t_{lag}$ | Time lag (time needed to reach the 63% of $J_{ss}$) |
| WE | Working electrode |

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
