# Peer review of "Hydrogen Diffusivity in Different Microstructures of 42CrMo4 Steel"

_hydrogen, doi:10.3390/hydrogen2040023_

Round 1

Reviewer 1 Report

I think the manuscript can be published as is.

Reviewer 2 Report

The manuscript describes the characterization of 42CrMo4 steel, treated at various conditions, to obtain a different structures that has an influence on trapping and diffusion of hydrogen atoms through the material, as this influence the phenomena known as hydrogen embrittlement.   

The experiments and presentation of results were done correctly, although some intervention is needed to obtain the manuscript more clearly.  

  1. Experimental Section: 

Table 2: QT600-7d must be explained in the footnote of Table 2. Does it mean 7 days?  

Section 2.2. Figure 1(a): 

From Figure 1(a) it is not clear, how it was obtained that working electrode be a cathode and an anode, at the same time, due to explicit claim in the manuscript (lines 137-138): "The oxidation or permeation current density, Jp , was continuously recorded in the anodic surface of the specimen by means of a pocketSTAT ….". 

Also, additional information about the anodic potential of oxidation (vs Ag/AgCl) of the Hdes should be provided.     

  1. Results:    

In Figure 6, it would be appropriate to insert values of the cathodic current, as it was performed in Figure 2(a).  

In Figures 7, 8, 9, ordinates have the value of 1012. It should be 10-12

Also, there is data for the current density highest of the 6 mA cm-2. Since it is in Experimantal Section presented that the overall current density was 6 mA cm-2, authors should adjust the final values of the x-axis.  

Also, the authors need to give attention to the inversion of the Dapp at the current density of 5 mA cm-2 in Figure 8. It was noticed in both treatments.  

Table 7. Although authors discussed concerning differences of the values of the Dapp obtained at tempered materials, concerning similar values of the DL, it would be good to present values of the tlag in Table 7.  

Reviewer 3 Report

The paper deals with the subject, which is relevant to advancements in several hydrogen-related areas, such as Hydrogen Storage, Hydrogen Applications, Hydrogen Transport and Infrastructure, as well as Hydrogen Safety. Accordingly, it is well suited to publication in the Hydrogen Journal.

The paper presents a well-done research, and it is also well organized and well written. Accordingly, I am delighted to recommend this work for publication. However, some things there do concern me, and I would kindly ask the authors to clear the issues, which are as follows.

As regards the first concern, I am not sure if this is a miswording, which is minor deficiency requiring some text editing, or a misconception, which would require pertinent explications and substantiations, i.e., a major revision of the paper. The matter is that several passages of the manuscript assert the causality (cause-and-effect) relationship between the hardness of a material (as the cause) and the hydrogen diffusivity in a material (as the consequence):

* “the effect of ... hardness on hydrogen transport” (lines 68-69);

* “the influence of the steel hardness ... in hydrogen diffusivity” (lines 71-72);

* “The influence of the hardness ... on both the apparent and the lattice hydrogen diffusion coefficients was assessed (lines 71-72)” (in the passage emphasizing the paper main achievement);

* “... microstructures ... always showed higher hydrogen diffusion coefficients ... due to their lower hardness and thus dislocation density.” (lines 260-262) (this, in addition, implies (“and thus”) that lower hardness is the cause of lower dislocation density, which, I’d say, is questionable)

* “… hardness determines the hydrogen diffusion coefficient…” (line 282);

“Dapp and DL values are strongly related with the hardness of the steel, having the kind of microstructure a much lower influence.” (in Conclusions, lines 328-329).

That is, taking the word meanings according, e.g., to the Oxford Dictionary, which are

Effect: change which is a result or consequence of an action or other cause;

Influence: the capacity to have an effect; synonym: effect,

do you really mean that the occurrence of a change of hardness per se has the ability to bring into existence the alteration of diffusivity? Do you mean that the diffusivity is some way deducible from the hardness like it is deducible from lattice vibrations, lattice potential energy relief, lattice spacings (geometry), etc. If so, such a concept requires substantiation clarifying the physics of supposed cause-consequence linkage. Though, this concept seems to be hardly defensible. On the other hand, I have no doubts that there exists a correlation between the diffusivity and the hardness, which both have the common cause – the microstructure. The paper convincingly shows this correlation. However, a correlation between two variables does not imply causation, and this latter requires pertinent proofs.

Another issue is about the implemented approach to evaluation of hydrogen diffusivities from step-charging experiments. Specifically, the chosen experimentation and data analysis procedure follows the recommendations of the ASTM G-148 Standard. This norm contains a warning as regards the derivation of diffusivities from step charging experiments because of the existence of initial hydrogen concentration in metal, etc., and it points out that more extensive analysis may be then required. More extensive comments about the matter would be appreciated.

Finally, I detected some language deficiencies and faults, such as, for example

- “Thomas and Szpunar [31] observed a decrease of the diffusion coefficient and on [¿] hydrogen trapping with the increase of the grain size...” (lines 76-77);

- “… have been performed in this study in order to study…” (lines 86-87)

- “… the steel microstructure in quenched and tempered steel...” (line 280).

Accordingly, it is advisable to perform careful checking of grammar, syntax and style.

Round 2

Reviewer 3 Report

Upon the paper revision, the authors did not convince me about the existence of not a correlation, but of a physical causality between the hardness and the hydrogen diffusivity, specifically, that the diffusivity decrease ineluctably results from the hardness increase as its inevitable consequence (cum hoc ergo propter hoc). Nevertheless, this is more about the interpretation of the work results than about the results per se. The results are valuable, and the paper advances the knowledge about metal-hydrogen interactions. The manuscript has sufficient merits to warrant publication.